# Repeated rebiopsy for detection of EGFR T790M mutation in patients with advanced-stage lung adenocarcinoma: Associated factors and treatment outcomes of Osimertinib

**Taeyun Kim[1,2☯], Junsu Choe[1☯], Sun Hye Shin[1], Byeong-Ho Jeong[1], Kyungjong Lee[1], Hojoong Kim[1], Se-Hoon Lee[3], Sang-Won Um[1,4]***

1 Department of Medicine, Division of Pulmonary and Critical Care Medicine, Samsung Medical Center, Sungkyunkwan University School of Medicine, Seoul, Republic of Korea, 2 Department of Medicine, Kosin University Gospel Hospital, Kosin University College of Medicine, Busan, Republic of Korea, 3 Department of Medicine, Division of Hematology and Oncology, Samsung Medical Center, Sungkyunkwan University School of Medicine, Seoul, Republic of Korea, 4 Department of Health Sciences and Technology, SAIHST, Sungkyunkwan University, Seoul, Republic of Korea

☯ These authors contributed equally to this work.
* sangwonum@skku.edu

**Data Availability Statement:** All relevant data are within the article and its Supporting Information files.

## Abstract

### Objectives

This study was performed to investigate the detection rate of EGFR T790M mutation by repeated rebiopsy, to identify the clinical factors related to repeated rebiopsy, and to assess survival outcomes according to the methods and numbers of repeated rebiopsies in patients with lung adenocarcinoma who received sequential osimertinib after failure of previous 1st or 2nd generation EGFR-tyrosine kinase inhibitors.

### Methods

This retrospective study included patients with advanced-stage lung adenocarcinoma who were confirmed to have EGFR T790M mutation and to have received osimertinib from January 2020 to February 2021 at Samsung Medical Center. The presence of T790M mutation was assessed based on either plasma circulating tumor DNA (ctDNA) or tissue specimens.

### Results

A total of 443 patients underwent rebiopsy, with 186 (42.0%) testing positive for the T790M mutation by the sixth rebiopsy. The final analysis included 143 eligible patients. Progression-free survival was not significantly different in terms of the methods (tissue: 13.3 months, 95% confidence interval [CI]: [9.4, 23.5] vs plasma: 11.1 months, 95% CI: [8.1, 19.4], p = 0.33) and numbers (one: 13.4 months, 95% CI: [9.4, 23.5] vs two or more: 11.0 months, 95% CI: [8.1, 14.8], p = 0.51) of repeated rebiopsies. Longer overall survival (OS) was found in patients in whom T790M was detected by tissue specimens rather than by plasma ctDNA (2-year OS rate: 81.7% for tissue vs 63.9% for plasma, p = 0.0038). Factors

**Funding:** This work was supported by the National Research Foundation of Korea (NRF) grant funded by the Korea government (MSIT) (2020R1A2C2006282). The funders had no role in study design, data collection and analysis, decision to publish, or preparation of the manuscript.

**Competing interests:** The authors have declared that no competing interests exist.

related to the lower numbers of rebiopsies included age and bone metastasis. Factor associated with T790M detection in tissue rather than in plasma was pleural metastasis, while advanced tumor stage was related to T790M confirmation in plasma rather than in tissue.

## Conclusions

Repeated rebiopsy for T790M detection in patients with NSCLC can increase the detection rate of the mutation. Detection of T790M by plasma ctDNA might be related to poor survival outcomes.

## Introduction

In patients with non-small cell lung cancer (NSCLC) with epidermal growth factor receptor (*EGFR*) mutation, EGFR tyrosine kinase inhibitors (TKIs) yield significant treatment responses. However, most cases progress after initial tumor shrinkage, and secondary resistance to EGFR-TKIs unavoidably leads to treatment failure [1]. T790M mutation is detected in approximately 50% of patients with NSCLC who develop resistance to EGFR-TKIs [2]. Osimertinib is effective against the T790M mutation and markedly improves survival outcomes for patients with advanced-stage EGFR-mutated NSCLC, especially for lung adenocarcinoma (LUAD), in whom previous regimens have failed [3].

Identification of the T790M mutation can guide physicians to properly initiate osimertinib or next-generation EGFR-TKIs for patients whose disease has progressed. At first rebiopsy, however, only 44% of patients were found to have the T790M mutation [4]. This may lead to subsequent rebiopsy attempts, which may in turn increase the detection rate of T790M for the prescription of osimertinib [5,6]. Therefore, it is necessary to attempt repeated rebiopsy to confirm the T790M mutation. However, few data exist regarding whether the T790M detection rate can be enhanced by repeated rebiopsy, which factors are associated with repeated rebiopsy, or survival outcomes in relation to these attempts. A recent meta-analysis of eight studies involving 1,013 patients with NSCLC revealed an increased T790M detection rate by repeated rebiopsy and no difference in progression-free survival (PFS) between first and repeated rebiopsy [6]. However, considerable heterogeneity was present among the studies because of the small numbers of participants and non-uniform measurement of *EGFR* mutation across different clinics [6].

In this context, the present study was performed to investigate the detection rate of T790M by repeated rebiopsy, identify the clinical factors related to repeated rebiopsy, and assess survival outcomes according to the methods and number of repeated rebiopsies in patients with lung adenocarcinoma who received sequential osimertinib after failure of previous 1st or 2nd generation EGFR-TKIs.

## Materials and methods

### Datasets and patient selection

This retrospective single-center observational study was performed at Samsung Medical Center. The electronic medical records of patients who met the following selection criteria from January 2020 to February 2021 were reviewed: aged ≥19 years; pathologically confirmed LUAD; advanced stage of disease, i.e., American Joint Committee on Cancer (AJCC) stage 3, 4A, or 4B; and the patients who received rebiopsy for the detection of T790M mutation.

The study protocol was approved by the Institutional Review Board of Samsung Medical Center (no. 2023-05-057). The study was conducted in accordance with the Declaration of Helsinki. All procedures were performed in accordance with relevant guidelines and regulations.

## Measurements

Information was collected on the patients' baseline characteristics, including age, sex, body mass index, smoking status (never or ever), tumor stage based on the 8th edition of the AJCC staging manual, and the Charlson comorbidity index. We also collected data on the number of metastatic organs, presence of metastasis to specific organs (brain, liver, bone, lung, or pleura), previous curative surgical treatment or radiotherapy, and the number of regimens before osimertinib. We collected data from the point at which treatment commenced in the palliative setting.

Tissue, plasma, and body fluid specimens were obtained for *EGFR* mutation analysis. For tissue specimens, *EGFR* mutations were confirmed using either a peptide nucleic acid clamp kit and real-time polymerase chain reaction (Roche Cobas *EGFR* mutation test; Roche Molecular Systems, Pleasanton, CA, USA) or next-generation sequencing (NGS) using a customized targeted panel (TruSight Oncology 500 Assay; Illumina, San Diego, CA, USA). Plasma circulating tumor DNA (ctDNA) was analyzed using either real-time polymerase chain reaction (Roche Cobas *EGFR* mutation test) or NGS (TruSight Oncology 500 Assay; Illumina, San Diego, CA, USA). Meanwhile, the next target for rebiopsy was based on the clinician's decision.

## Outcomes

PFS with osimertinib and overall survival (OS) were measured. PFS was defined as the time from initiation of osimertinib to tumor progression or death during osimertinib therapy. OS was defined as the time from the initiation of osimertinib to death of any cause. Data from patients who were still receiving treatment or were alive at the time of data collection were censored (31 May 2023).

## Statistical analysis

Data are presented as mean ± standard deviation or median (interquartile range: IQR) for continuous variables and n (%) for categorical variables. Exploratory data analyses were performed to identify the impact of patients' baseline characteristics. Student's t-test and the chi-square test or Fisher's exact test were used to compare the methods and number of repeated rebiopsies according to the patients' characteristics for continuous and categorical variables, respectively. The Kaplan–Meier method and log-rank test were performed to visualize and compare PFS and OS according to the methods and number of rebiopsies for T790M confirmation. Subgroup analyses were performed by sex, age, smoking status, and AJCC stage. A multivariable logistic regression model with a stepwise selection process was used to identify factors related to repeated rebiopsy. P-value below 0.05 was considered statistically significant. All statistical analyses were performed using R software (version 4.3.3 for Windows; R Development Core Team).

## Results

### Baseline characteristics

**Fig 1** shows the patient selection process. A total of 443 patients with LUAD received rebiopsies, and 186 were positive for T790M after sequential rebiopsies (**Table 1**), with the

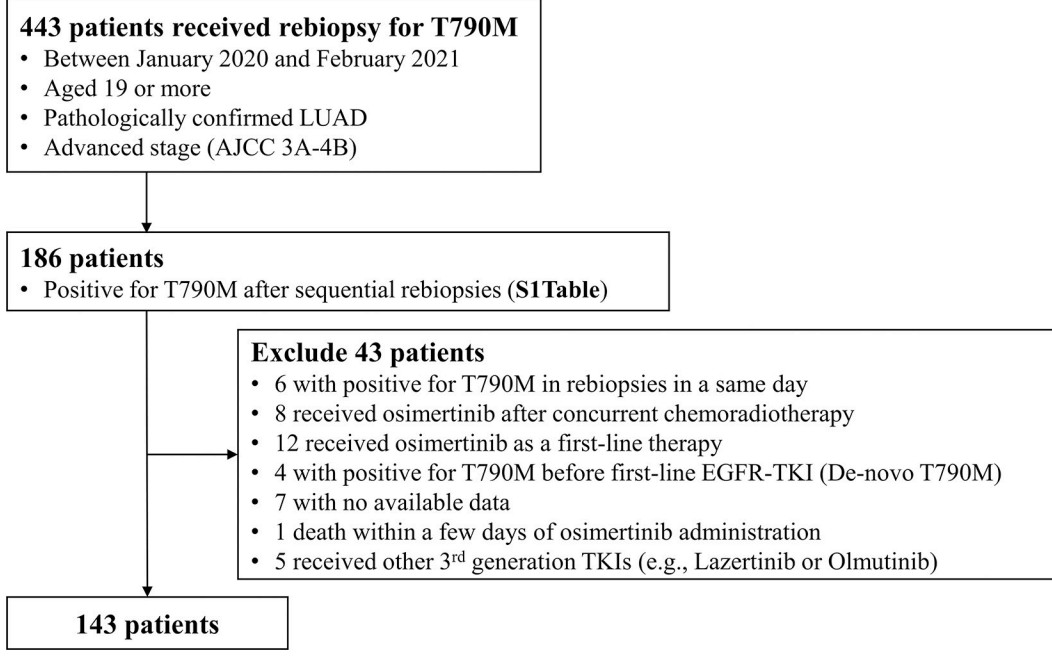

**Fig 1. Patient selection process.** LUAD, lung adenocarcinoma; AJCC, American Joint Committee on Cancer; EGFR, epidermal growth factor receptor; TKI, tyrosine kinase inhibitor.

cumulative rate of T790M detection increasing as follows: 14.2% after the first rebiopsy, 31.8% after the second, 38.8% after the third, 41.1% after the fourth, 41.5% after the fifth, and 42.0% after the sixth. Because this study was designed to analyze patients with LUAD who were confirmed to have T790M mutation and to have received sequential osimertinib in a palliative setting, we excluded 43 patients for the following reasons: underwent rebiopsy on the same day as positive confirmation of T790M (n = 6), treatment with osimertinib after concurrent radiotherapy (n = 8), treatment with osimertinib as first-line therapy (n = 12), positivity for T790M before first-line therapy (e.g., *de novo* T790M mutation) (n = 4), no available data (n = 7), death within a few days of osimertinib initiation (n = 1), and received other 3rd generation TKIs (e.g., lazertinib or olmutinib). For the final analysis, 143 patients were included.

Clinical characteristics of 143 study subjects are displayed in Tables 2 and S1. The patients' mean age was 66 years, and 55.9% were female (**Table 2**). Stage 4B cancer was present in 52.4% of the patients. Of all 143 patients, the most prevalent site of metastasis was the lung (n = 71, 49.7%), followed by the pleura (n = 67, 46.9%), bone (n = 55, 38.5%), brain (n = 50, 35.0%), and liver (n = 18, 12.6%). Curative chemoradiotherapy had been performed previously

**Table 1. Detection rate of T790M in all patients who received a rebiopsy during the study period.**

|  | First | Second | Third | Fourth | Fifth | Sixth |
|---|---|---|---|---|---|---|
| **T790M, n (%)** |  |  |  |  |  |  |
| Positive | 63 (14.2) | 78 (21.5) | 31 (16.1) | 10 (10.1) | 2 (4.8) | 2 (12.5) |
| Negative | 380 (85.8) | 285 (78.5) | 162 (83.9) | 89 (89.9) | 40 (95.2) | 14 (87.5) |
| **Total** | 443 | 363 | 193 | 99 | 42 | 16 |
| Liquid, n (%) | 330 (74.5) | 148 (40.8) | 80 (41.5) | 43 (43.4) | 18 (42.9) | 5 (31.3) |
| Tissue, n (%) | 113 (25.5) | 215 (59.2) | 113 (58.5) | 56 (56.6) | 24 (57.1) | 11 (68.8) |
| **Cumulative % for T790M** | 14.2 | 31.8 | 38.8 | 41.1 | 41.5 | 42.0 |

**Table 2. Patients' baseline characteristics (N = 143).**

| | |
|---|---|
| **Age, years** | 66.0 ± 10.2 |
| **Female sex** | 80 (55.9) |
| **Body mass index, kg/m$^2$** | 23.3 ± 3.5 |
| **Smoking status** | |
| Never | 91 (63.6) |
| Ever | 52 (36.4) |
| **Charlson comorbidity index** | |
| 6 | 119 (83.2) |
| ≥ 7 | 24 (16.8) |
| **AJCC tumor stage** | |
| 3–4A | 68 (47.5) |
| 4B | 75 (52.4) |
| **Number of metastatic organs** | |
| 0–1 | 56 (39.2) |
| 2–3 | 68 (47.6) |
| ≥ 4 | 19 (13.3) |
| *EGFR* **mutation** | |
| Exon 19 Deletion | 88 (61.5) |
| *L858R* | 52 (36.4) |
| Others | 3 (2.1) |
| **Brain metastasis** | 50 (35.0) |
| **Liver metastasis** | 18 (12.6) |
| **Bone metastasis** | 55 (38.5) |
| **Lung-to-lung metastasis** | 71 (49.7) |
| **Pleural metastasis** | 67 (46.9) |
| **Previous curative chemoradiotherapy** | 20 (14.0) |
| **Previous curative surgery** | 35 (24.5) |
| **Previous EGFR-TKI treatment** Gefitinib Afatinib | 68 (47.6) 62 (43.4) |
| Erlotinib | 13 (9.1) |
| **Number of previous regimens** | |
| 1 | 108 (75.5) |
| 2 | 21 (14.7) |
| 3 | 14 (9.8) |

Data are presented as mean ± standard deviation for continuous variables and n (%) for categorical variables.

AJCC, American Joint Committee on Cancer.

in 20 (14.0%) patients, and a curative operation had been previously performed in 35 (24.5%) patients. Exon 19 deletion was found in 88 (61.5%) patients, which was higher than the rate of L858R (52, 36.4%). Most of patients received gefitinib (68, 47.6%) or afatinib (62, 43.4%) before the initiation of osimertinib.

## Clinical outcomes and survival

The median interval between the first and second rebiopsy was 12.5 days (IQR: 8.5–21.0 days). PFS was not significantly different in terms of the methods (tissue: 13.3 months, 95% confidence interval [CI]: [9.4, 23.5] vs plasma: 11.1 months, 95% CI: [8.1, 19.4], p = 0.33)

**Table 3. Progression-free survival (months) according to the number of rebiopsies and methods for confirmation of T790M mutation.**

|  | 1-year PFS rate | Median | 95% CI | p |
|---|---|---|---|---|
| **Number of rebiopsies** |  |  |  | 0.51 |
| One | 54.7% | 13.4 | 10.3–20.5 |  |
| Two or more | 40.9% | 11.0 | 8.1–14.8 |  |
| **Confirmation of T790M** |  |  |  | 0.33 |
| Tissue | 55.2% | 13.3 | 9.4–23.5 |  |
| Plasma | 46.9% | 11.1 | 8.1–19.4 |  |
| **Confirmation of T790M** |  |  |  | 0.49 |
| Tissue, first | 62.2% | 16.6 | 12.0–28.8 |  |
| Plasma, first | 50.3% | 12.2 | 8.3–19.6 |  |
| Tissue, second or more | 47.6% | 11.2 | 8.1–26.4 |  |
| Plasma, second or more | 16.7% | 9.0 | 5.6–NA |  |

PFS, progression free survival; CI, confidence interval; NA, not available.

and numbers (one: 13.4 months, 95% CI: [9.4, 23.5] vs two or more: 11.0 months, 95% CI: [8.1, 14.8], p = 0.51) of repeated rebiopsies (**Table 3**). When confining the analysis to patients whose tumor stage was > 3A at the initial diagnosis, the result was not different (**S2 Table**).

In terms of OS, no significant difference according to the number of rebiopsies was found (**Table 4**). However, when patients were stratified by the method used to confirm T790M, a significant difference in OS was observed (**Fig 2**): patients whose mutation was confirmed through a tissue specimen survived for a longer period than patients whose mutation was detected in a plasma sample (i.e., ctDNA) (2-year OS rate: 81.7% for tissue vs 63.9% for plasma, p = 0.0038). A similar trend was found when further stratifying patients by both the method and number of repeated rebiopsies. When confining the analysis to patients with an advanced tumor stage at the initial diagnosis, it was also found that the OS was different between the methods of T790M confirmation (tissue vs. plasma) (**S1 Fig** and **S3 Table**).

**Table 4. Overall survival (months) according to the number of rebiopsies and methods for confirmation of T790M mutation.**

|  | 2-year OS rate | Median | 95% CI | p |
|---|---|---|---|---|
| **Number of rebiopsies** |  |  |  | 0.13 |
| One | 70.6% | NR | 32.2–NA |  |
| Two or more | 78.3% | NR | NA–NA |  |
| **Confirmation of T790M** |  |  |  | 0.0038 |
| Tissue | 81.7% | NR | NA–NA |  |
| Plasma | 63.9% | NR | 26.3–NA |  |
| **Confirmation of T790M** |  |  |  | 0.031 |
| Tissue, first | 79.9% | NR | NA–NA |  |
| Plasma, first | 64.7% | NR | 27.8–NA |  |
| Tissue, second or more | 84.1% | NR | NA–NA |  |
| Plasma, second or more | 53.3% | NR | 14.2–NA |  |

OS, overall survival; CI, confidence interval; NA, not available; NR, not reached.

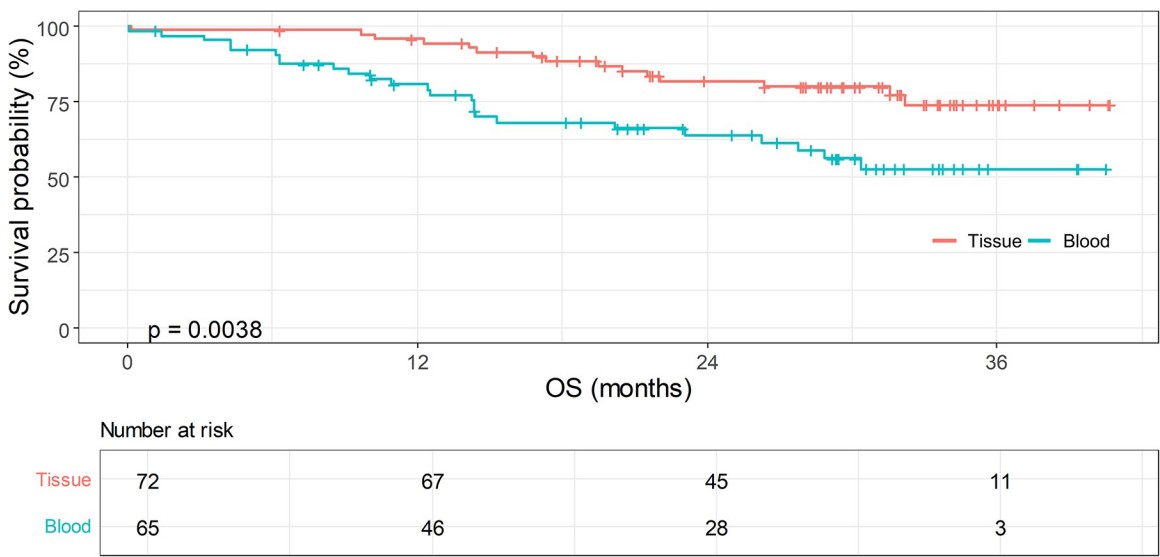

**Fig 2. Difference in OS between the methods of T790M confirmation.** OS, overall survival.

## Clinical characteristics associated with the number of rebiopsies

The multivariable-adjusted odds ratios of factors influencing the number of rebiopsies are shown in **Table 5**. Older age and bone metastasis were significantly associated with a decreased likelihood of undergoing a second or subsequent rebiopsies.

## Clinical characteristics associated with detection of T790M in plasma ctDNA

The characteristics of patients according to the sites of T790M detection are summarized in **S4 Table**. Stage 4B, brain metastasis, bone metastasis and pleural metastasis were more common

**Table 5. Factors associating influencing the number of rebiopsies and the confirmation of T790M by plasma ctDNA analysis.**

|  | OR* | 95% CI | p |
|---|---|---|---|
| **2 or more rebiopsies** |  |  |  |
| Age | 0.94 | 0.90–0.98 | 0.006 |
| Bone metastasis | 0.19 | 0.06–0.46 | <0.001 |
| Number of previous regimens |  |  |  |
| ≥ 2 (reference: 1) | 0.39 | 0.13–1.09 | 0.087 |
| **T790M confirmation by plasma ctDNA analysis** |  |  |  |
| Age | 1.03 | 0.99–1.08 | 0.124 |
| Stage 4B (reference: 3–4A) | 3.81 | 1.70–8.87 | 0.001 |
| Pleural metastasis | 0.44 | 0.19–0.99 | 0.048 |
| Liver metastasis | 2.68 | 0.76–11.26 | 0.144 |
| Number of previous treatment regimens | 3.91 | 1.00–19.34 | 0.064 |
| ≥ 2 (reference: 1) | 1.96 | 0.77–5.12 | 0.16 |

* OR was calculated using a stepwise logistic regression model considering age, sex, body mass index, smoking status, *EGFR* mutation status, Charlson's comorbidity index, tumor stage, numbers of organ metastasis, presence of brain, liver, bone, and pleural metastasis, and the numbers of previous regimens.

OR of > 1 indicates a higher probability of undergoing multiple rebiopsies or detecting T790M by plasma.

OR, odds ratio; ctDNA, circulating tumor DNA.

in the group with T7900M confirmation using plasma ctDNA compared to those with T790M confirmation using tissue biopsy. The multivariable-adjusted odds ratios of T790M confirmation through plasma ctDNA are presented in **Table 5**. Pleural metastasis was related to mutation detection in tissue specimens. Conversely, more advanced tumor stage (4B) was related to the detection in plasma samples.

## Discussion

In this single-institution retrospective study, we analyzed patients with advanced-stage LUAD who underwent failed EGFR-TKI therapy and other therapies and were subsequently treated with osimertinib in the presence of T790M mutation. We comprehensively followed the trajectories of T790M confirmation according to the number of rebiopsies and the site of mutation detection. The detection rate of T790M detection rate in all patients who received sequential rebiopsies was 42.0%. OS were longer in patients whose mutation was detected in tissue than in plasma: it was similar in the analysis that confined to the patients who were initially diagnosed at an advanced stage and had received no previous curative surgery or chemoradiotherapy. Specifically, patients whose mutation was initially found in tissue specimens had the longest survival, followed by those whose mutation was found in tissue specimens at the second or more rebiopsy, plasma samples at the first rebiopsy, and plasma samples at the second or more rebiopsy. Factors associated with the number of rebiopsies included older age and bone metastasis. Factor associated with T790M confirmation in tissue rather than in plasma was the presence of pleural metastasis, while advanced tumor stage was related to T790M confirmation in plasma rather than in tissue.

OS was significantly longer when the T790M mutation was found in tissue than in plasma. This difference in survival might be attributed to the fact that patients whose mutation was detected in plasma samples had more advanced disease and a higher rate of metastasis to specific organs at the initiation of osimertinib (**S4 Table**). Although several prior studies have examined differences in survival outcomes (e.g., PFS or OS) based on the sites at which T790M mutation was confirmed, the results are inconsistent [5,7–12]. For example, some studies showed differences in PFS according to the rebiopsy site [7,9,10], whereas others showed no differences [8,12]. Relatively few studies have reported relevant data on OS, but they consistently showed poorer OS when the mutation was found in plasma rather than in tissue [10,11]. Our findings are in line with these previous results regarding OS.

The overall detection rate of T790M was comparable but slightly lower to previous reports [6,7,13–15]. In our study, the detection rate after first rebiopsy was 14.2%, 31.8% after the second, 38.8% after the third, 41.1% after the fourth, 41.5% after the fifth, and 42.0% after the sixth. A retrospective study of 80 patients with NSCLC showed that, in 53.7% of patients who progressed after first-line therapy with first- or second-generation T790M mutation in EGFR-TKIs was found at the first rebiopsy, and subsequent rebiopsies increased the detection rate by 12.5%, 2.5%, 1.2%, and 1.2% [14]. Another meta-analysis showed that the detection rate after the first rebiopsy was 44.2%, and the rate after repeated rebiopsy was 46.5%; the pooled detection rate of the first and repeated rebiopsies was 54.5% [6]. This discrepancy likely originated from the differences in the method by which the repeated rebiopsy performed: In our institution, liquid biopsy was used as the primary method for T790M detection (74.5%, 330/443), likely due to the simplicity of blood sampling compared to tissue acquisition. Meanwhile, tissue biopsy was the predominant method for the second rebiopsy (59.2%, 215/363). Interindividual spatiotemporal heterogeneity of *EGFR* mutations, especially T790M, has been demonstrated in patients with NSCLC for whom EGFR-TKI therapy failed [16,17]. This could result in an unsuccessful single attempt at the detection of T790M mutation, necessitating

additional attempts at different sites. Many advances in techniques have led to the use of liquid biopsy as a preferred method for T790M detection [18]. However, there still exists a group of patients in whom T790M is more likely to be confirmed within tissue specimens [19]. An individualized approach to detect T790M mutation is necessary.

With respect to establishing individualized strategies for T790M detection, our findings regarding the factors related to the number of rebiopsies and the sites of mutation detection add valuable information to the literature. Some studies have suggested potential factors related to positive T790M detection. A multicenter retrospective study in Italy showed that bone metastasis and high numbers of metastatic organs were associated with T790M detection [20]. Our study similarly indicated that patients with advanced disease at the time of rebiopsy and patients with long-standing disease may benefit from the less invasive method of ctDNA sampling to detect T790M. Meanwhile, a negative correlation was found between pleural metastasis and the possibility of detection by a plasma sample. This difference from other organs might be related to the tumor microenvironment, molecular pathways, or genetic traits that affect tumor affinity [21]. Data on the negative correlation of pleural metastasis with other sites of metastasis have accumulated from observational studies [22,23].

Our study has several limitations. First, it was a retrospective study based on electronic medical records of a single institution, and the results are not generalizable to all hospitals or clinical settings. For example, multiple rebiopsies might not be feasible in many institutions. Second, given the nature of the study design, causality in the relationships of associated factors with the number and methods of repeated rebiopsies should be inferred with caution. Third, because our institution initiated liquid biopsy in 2018, the follow-up period was relatively short. Further data after the survival period are necessary to elucidate the longitudinal impact of repeated rebiopsy.

## Conclusion

Repeated rebiopsy for the confirmation of T790M mutation in patients with LUAD who underwent treatment failure with previous regimens, including EGFR-TKI therapy, may increase the detection rate of T790M mutation. This could in turn guide physicians in terms of starting their patients on osimertinib. In particular, active rebiopsy of tissue could be considered in younger patients or those with a lower disease stage because there is a higher likelihood that mutations will not be detected in plasma samples. Confirmation of T790M in plasma rather than tissue may be related to poor survival outcomes, which might reflect advanced disease. Given the inter- and intraindividual spatiotemporal heterogeneity of *EGFR* mutations, it is important to personalize the strategy for repeated rebiopsy.

## Supporting information

**S1 Fig. Difference in OS between the methods of T790M confirmation in patients with advanced-stage disease at diagnosis.** OS, overall survival. *Patients who were diagnosed with stage 3, 4A, or 4B at their first diagnosis were analyzed.
(TIF)

**S1 Table. Clinical data set of 143 study subjects.**
(XLSX)

**S2 Table. Progression-free survival (months) according to the number of rebiopsies and methods for confirmation of T790M mutation in patients initially diagnosed with advanced disease.**
(DOCX)

**S3 Table. Overall survival (months) according to the number of rebiopsies and methods for confirmation of T790M mutation in patients initially diagnosed with advanced disease.** (DOCX)

**S4 Table. Patient characteristics according to sites of T790M confirmation.** (DOCX)

## Author Contributions

**Conceptualization:** Sang-Won Um.

**Data curation:** Taeyun Kim, Junsu Choe.

**Formal analysis:** Taeyun Kim, Sang-Won Um.

**Funding acquisition:** Sang-Won Um.

**Validation:** Junsu Choe, Sun Hye Shin, Byeong-Ho Jeong, Kyungjong Lee, Hojoong Kim, Se-Hoon Lee, Sang-Won Um.

**Writing – original draft:** Taeyun Kim.

**Writing – review & editing:** Sang-Won Um.

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
