## [Decision Letter · Decision Letter 0]

18 Jun 2024

PONE-D-24-12639Repeated rebiopsy for detection of EGFR T790M mutation in patients with advanced-stage lung adenocarcinoma: associated factors and treatment outcomes of OsimertinibPLOS ONE

Dear Dr. Um,

Thank you for submitting your manuscript to PLOS ONE. After careful consideration, we feel that it has merit but does not fully meet PLOS ONE’s publication criteria as it currently stands. Therefore, we invite you to submit a revised version of the manuscript that addresses the points raised during the review process.

We look forward to receiving your revised manuscript.

Kind regards,

Hamidreza Montazeri Aliabadi

Academic Editor

PLOS ONE

Journal Requirements:

"This work was supported by the National Research Foundation of Korea (NRF) grant funded by the Korea government (MSIT) (2020R1A2C2006282)."

3. In the online submission form, you indicated that "The datasets used and/or analyzed during the current study available from the corresponding author on reasonable request."

Reviewers' comments:

Reviewer's Responses to Questions

**Comments to the Author**

1. Is the manuscript technically sound, and do the data support the conclusions?

Reviewer #1: Yes

Reviewer #2: Partly

2. Has the statistical analysis been performed appropriately and rigorously? 

Reviewer #1: Yes

Reviewer #2: Yes

3. Have the authors made all data underlying the findings in their manuscript fully available?

Reviewer #1: Yes

Reviewer #2: Yes

4. Is the manuscript presented in an intelligible fashion and written in standard English?

Reviewer #1: Yes

Reviewer #2: Yes

5. Review Comments to the Author

Reviewer #1: This is an interesting study about the T790M yield of repeated biopsies under EGFR TKI progression and their association with patient outcomes.

I have the following comments:

1. It would have been helpful to give some information about the entire patient population: 100% had T790M after max 3 biopsies, but how many received biopsies (i.e. how many other patients had all biopsies negative) in the same time period?

2. It would be important to add the type of EGFR mutation in Table 1 (del19 vs. L858R vs. other rare mutations)

3. Regarding the analysis of survival: it would be important to also check other prognostic parameters, like the type of EGFR mutation. It is known that the type of EGFR mutation and presence of brain metastases are independent predictors of OS for these patients, which should be taken into account in the analysis, see for example https://pubmed.ncbi.nlm.nih.gov/32871455/, which could also be added in the Discussion.

Reviewer #2: Um et al reported implications of repeated rebiopsy for T790M detection in patients with EGFR-mutant NSCLC and associated factors and treatment outcomes of osimertinib.

Their manuscript includes several interesting points for readers. However, there is a critical methodologic problem in their study, and major modification is necessary.

Major points:

1. Their study population focused on only osimertinib-administered patients. They insisted T790M detection rate on first rebiopsy of 69.9%. This is absolutely incorrect. As they mentioned, many studies reported clinical T790M detection rate on first rebiopsy of 30-50%. These previous studies consecutively examined all patients who received rebiopsy to detect T790M in their hospitals. The authors should also consecutively examine all rebiopsied patients in their institute. This is a critical problem in their study.

2. Similarly, T790M detection rates on second and third rebiopsies were incorrect. They excluded cases without T790M, since their study only examined T790M-positive cases.

3. They described “The median interval between the first and second rebiopsy was 12.5 days (IQR: 8.5-21.0 days).” This is astonished. After failure of first rebiopsy, was second rebiopsy performed on the same target? or another target? Their manuscript did not show detailed information on rebiopsy targets.

Minor point:

1. EGFR should be italicized when they express a gene. (e.g., EGFR mutation)

6. PLOS authors have the option to publish the peer review history of their article (what does this mean?). If published, this will include your full peer review and any attached files.

Reviewer #1: **Yes: **Petros Christopoulos

Reviewer #2: No

---

## [Author Response · Author response to Decision Letter 0]

30 Jul 2024

Reviewer 1

Comment 1: It would have been helpful to give some information about the entire patient population: 100% had T790M after max 3 biopsies, but how many received biopsies (i.e. how many other patients had all biopsies negative) in the same time period?

Response 1: Thank you for the valuable comment. 

In response to the comments from both reviewers, we re-checked the entire patient population who received rebiopsies during the study period. Therefore, we have also revised the patient selection process. In brief, a total of 443 patients received rebiopsies, with the cumulative rate of T790M detection increasing as follows: 14.2% after the first rebiopsy, 31.8% after the second, 38.8% after the third, 41.1% after the fourth, 41.5% after the fifth, and 42.0% after the sixth.

We summarized these results as Fig 1 and Table 1. Also, we have revised the related sections: Abstract (Results), Materials and Methods, 1st paragraph of Results, and 1st and 3rd paragraph of Discussion.

Comment 2 & 3: It would be important to add the type of EGFR mutation in Table 1 (del19 vs. L858R vs. other rare mutations). Regarding the analysis of survival: it would be important to also check other prognostic parameters, like the type of EGFR mutation. It is known that the type of EGFR mutation and presence of brain metastases are independent predictors of OS for these patients, which should be taken into account in the analysis, see for example https://pubmed.ncbi.nlm.nih.gov/32871455/, which could also be added in the Discussion.

Response 2 & 3: Thank you for the comment and sharing a valuable reference. 

First, we have added numbers and percentages of EGFR mutation in Table 2. Second, we have re-estimated odds ratio from multivariable analyses additionally considering EGFR mutation status. Results are summarized in Table 5: however, in this analysis we could not find EGFR mutation as a distinguishing parameter in terms of number of rebiopsy and method for rebiopsy. Please refer to the Results section. 

Reviewer 2

Major points:

Comment 1 & 2: Their study population focused on only osimertinib-administered patients. They insisted T790M detection rate on first rebiopsy of 69.9%. This is absolutely incorrect. As they mentioned, many studies reported clinical T790M detection rate on first rebiopsy of 30-50%. These previous studies consecutively examined all patients who received rebiopsy to detect T790M in their hospitals. The authors should also consecutively examine all rebiopsied patients in their institute. This is a critical problem in their study.

Similarly, T790M detection rates on second and third rebiopsies were incorrect. They excluded cases without T790M, since their study only examined T790M-positive cases.

Response 1 & 2: Thank you for the valuable comment. 

In response to the comments from both reviewers, we have re-checked the entire patient population who received rebiopsies during the study period. Therefore, we have also revised the patient selection process. In brief, a total of 443 patients received rebiopsies, with the cumulative rate of T790M detection increasing as follows: 14.2% after the first rebiopsy, 31.8% after the second, 38.8% after the third, 41.1% after the fourth, 41.5% after the fifth, and 42.0% after the sixth.

We have summarized these results as Fig 1 and Table 1. Also, we have revised the related sections: Abstract (Results), Materials and Methods, 1st paragraph of Results, and 1st and 3rd paragraph of Discussion.

Comment 3: They described “The median interval between the first and second rebiopsy was 12.5 days (IQR: 8.5-21.0 days).” This is astonished. After failure of first rebiopsy, was second rebiopsy performed on the same target? or another target? Their manuscript did not show detailed information on rebiopsy targets.

Response 3: Thank you for the comment. The next target for rebiopsy was based on the clinician’s decision rather than on a protocolized approach. Therefore, we clarified this point in the method section. 

Minor point:

Comment 1: EGFR should be italicized when they express a gene. (e.g., EGFR mutation)

Response 1: We have italicized EGFR when it expresses the gene.

---

## [Decision Letter · Decision Letter 1]

16 Aug 2024

PONE-D-24-12639R1Repeated rebiopsy for detection of EGFR T790M mutation in patients with advanced-stage lung adenocarcinoma: associated factors and treatment outcomes of OsimertinibPLOS ONE

Dear Dr. Um,

Thank you for submitting your manuscript to PLOS ONE. After careful consideration, we feel that it has merit but does not fully meet PLOS ONE’s publication criteria as it currently stands. Therefore, we invite you to submit a revised version of the manuscript that addresses the points raised during the review process. **Please make sure to specifically address the concerns raised by Reviewer 2.**

We look forward to receiving your revised manuscript.

Kind regards,

Hamidreza Montazeri Aliabadi

Academic Editor

PLOS ONE

Journal Requirements:

Reviewers' comments:

Reviewer's Responses to Questions

**Comments to the Author**

1. If the authors have adequately addressed your comments raised in a previous round of review and you feel that this manuscript is now acceptable for publication, you may indicate that here to bypass the “Comments to the Author” section, enter your conflict of interest statement in the “Confidential to Editor” section, and submit your "Accept" recommendation.

Reviewer #1: All comments have been addressed

Reviewer #2: All comments have been addressed

2. Is the manuscript technically sound, and do the data support the conclusions?

Reviewer #1: Yes

Reviewer #2: Yes

3. Has the statistical analysis been performed appropriately and rigorously? 

Reviewer #1: Yes

Reviewer #2: Yes

4. Have the authors made all data underlying the findings in their manuscript fully available?

Reviewer #1: Yes

Reviewer #2: Yes

5. Is the manuscript presented in an intelligible fashion and written in standard English?

Reviewer #1: Yes

Reviewer #2: Yes

6. Review Comments to the Author

**Reviewer #1:** The authors have addressed all reviewers' comments and improved the manuscript accordingly.

There are no further comments

**Reviewer #2:** Um et al revised and resubmitted their manuscript with major modifications.

Although their manuscript has several interesting and clinically meaningful points, there are still some unresolved critical problems in the manuscript.

I feel Figure 1 was well modified and easy to understand for readers and their conclusions are excellent.

Major points:

1. Results of abstract is inappropriate for a scientific paper. They exhibited little numeric data. Guide for authors of PLOS ONE describes, “Results – What did you find? Briefly give the key findings of your study. Include key numeric data (including confidence intervals or p values), where possible.”. They should thoroughly modify it. Table 1 is impressive and easy to understand for readers, and the findings should be included in the Results section.

2. S3 Table appears first in the Discussion. Also, sentences on lines 256-259: “In our institution, liquid biopsy was used as the primary method for T790M detection (73.7%, 330/448), likely due to the simplicity of blood sampling compared to tissue acquisition. Meanwhile, tissue biopsy was the predominant method for the second rebiopsy (59.2%, 215/363).” appear first in the Discussion. Such expressions are really inappropriate for scientific papers. What the authors did not mention in the Result should not be used in the Discussion. They must mention these results in the Result section, previously.

3. T790M-positive rate at first rebiopsy was only 14%. This is lower than former pivotal reports on T790M-positive rate: 30-50%. They considered high rate of liquid biopsy caused this lower T790M-positive rate. So, they should demonstrate rates of liquid and histologic biopsies from first to sixth rebiopsy in the Result section.

4. On lines 146-147, they described, “For the final analysis, 143 patients were included and they received a maximum of three repeated rebiopsies.”. This is confusing. Table 1 shows some patients received sixth rebiopsy. What?

7. PLOS authors have the option to publish the peer review history of their article (what does this mean?). If published, this will include your full peer review and any attached files.

Reviewer #1: **Yes: **Petros Christopoulos

Reviewer #2: No

---

## [Author Response · Author response to Decision Letter 1]

19 Aug 2024

Response to Reviewer’s comments

Um et al revised and resubmitted their manuscript with major modifications.

Although their manuscript has several interesting and clinically meaningful points, there are still some unresolved critical problems in the manuscript.

I feel Figure 1 was well modified and easy to understand for readers and their conclusions are excellent.

Major points:

Comment 1: Results of abstract is inappropriate for a scientific paper. They exhibited little numeric data. Guide for authors of PLOS ONE describes, “Results – What did you find? Briefly give the key findings of your study. Include key numeric data (including confidence intervals or p values), where possible.”. They should thoroughly modify it. Table 1 is impressive and easy to understand for readers, and the findings should be included in the Results section.

Response 1: Thank you for the comment. We have added key numerical data with p values to the Results section of Abstract as follows. 

“Progression-free survival was not significantly different in terms of the methods (tissue: 13.3 months, 95% confidence interval [CI]: [9.4, 23.5] vs plasma: 11.1 months, 95% CI: [8.1, 19.4], p=0.33) and numbers (one: 13.4 months, 95% CI: [9.4, 23.5] vs two or more: 11.0 months, 95% CI: [8.1, 14.8], p=0.51) of repeated rebiopsies. Longer overall survival (OS) was found in patients in whom T790M was detected by tissue specimens rather than by plasma ctDNA (2-year OS rate: 81.7% for tissue vs 63.9% for plasma, p = 0.0038).”

We have also added the key numerical data in the Results section.

Comment 2: S3 Table appears first in the Discussion. Also, sentences on lines 256-259: “In our institution, liquid biopsy was used as the primary method for T790M detection (73.7%, 330/448), likely due to the simplicity of blood sampling compared to tissue acquisition. Meanwhile, tissue biopsy was the predominant method for the second rebiopsy (59.2%, 215/363).” appear first in the Discussion. Such expressions are really inappropriate for scientific papers. What the authors did not mention in the Result should not be used in the Discussion. They must mention these results in the Result section, previously.

Response 2: Thank you for your comments. 

We have added following sentences in the Results section (Clinical characteristics associated with detection of T790M in plasma ctDNA).

“The characteristics of patients according to the sites of T790M detection are summarized in S3 Table. Stage 4B, brain metastasis, bone metastasis and pleural metastasis were more common in the group with T7900M confirmation using plasma ctDNA compared to those with T790M confirmation using tissue biopsy.”

We have also added following sentences in the Results section (Baseline Characteristics).

“The majority of patients (74.5%) received their first rebiopsy using a liquid biopsy, while most patients at the second rebiopsy received a tissue biopsy (59.2%).”

Comment 3: T790M-positive rate at first rebiopsy was only 14%. This is lower than former pivotal reports on T790M-positive rate: 30-50%. They considered high rate of liquid biopsy caused this lower T790M-positive rate. So, they should demonstrate rates of liquid and histologic biopsies from first to sixth rebiopsy in the Result section.

Response 3: Thank you for your insightful comment. We have added the data on the rates of liquid and tissue biopsies from first to sixth rebiopsy in Table 1 and have described the content in the Results section

Comment 4: On lines 146-147, they described, “For the final analysis, 143 patients were included and they received a maximum of three repeated rebiopsies.”. This is confusing. Table 1 shows some patients received sixth rebiopsy. What?

Response 4: We apologize for the confusion. To be clear, we have revised the sentence as follows: “For the final analysis, 143 patients were included”

---

## [Decision Letter · Decision Letter 2]

26 Aug 2024

Repeated rebiopsy for detection of EGFR T790M mutation in patients with advanced-stage lung adenocarcinoma: associated factors and treatment outcomes of Osimertinib

PONE-D-24-12639R2

Dear Dr. Um,

We’re pleased to inform you that your manuscript has been judged scientifically suitable for publication and will be formally accepted for publication once it meets all outstanding technical requirements.

Kind regards,

Hamidreza Montazeri Aliabadi

Academic Editor

PLOS ONE

Additional Editor Comments (optional):

Reviewers' comments:

Reviewer's Responses to Questions

**Comments to the Author**

1. If the authors have adequately addressed your comments raised in a previous round of review and you feel that this manuscript is now acceptable for publication, you may indicate that here to bypass the “Comments to the Author” section, enter your conflict of interest statement in the “Confidential to Editor” section, and submit your "Accept" recommendation.

Reviewer #2: All comments have been addressed

2. Is the manuscript technically sound, and do the data support the conclusions?

Reviewer #2: Yes

3. Has the statistical analysis been performed appropriately and rigorously? 

Reviewer #2: Yes

4. Have the authors made all data underlying the findings in their manuscript fully available?

Reviewer #2: Yes

5. Is the manuscript presented in an intelligible fashion and written in standard English?

Reviewer #2: Yes

6. Review Comments to the Author

Reviewer #2: (No Response)

7. PLOS authors have the option to publish the peer review history of their article (what does this mean?). If published, this will include your full peer review and any attached files.

Reviewer #2: No

---

## [Editor Report · Acceptance letter]

10 Sep 2024

PONE-D-24-12639R2 

PLOS ONE

Dear Dr. Um, 

I'm pleased to inform you that your manuscript has been deemed suitable for publication in PLOS ONE. Congratulations! Your manuscript is now being handed over to our production team.

Kind regards, 

on behalf of

Dr. Hamidreza Montazeri Aliabadi 

Academic Editor

PLOS ONE